# A Two-Stage Gradient Ascent-Based Superpixel Framework for Adaptive Segmentation

**Wangpeng He ***[ID]**, Cheng Li, Yanzong Guo, Zhifei Wei and Baolong Guo**

School of Aerospace Science and Technology, Xidian University, Xi'an 710071, China;
licheng812@stu.xidian.edu.cn (C.L.); imoldpan@gmail.com (Y.G.); weizhifei@stu.xidian.edu.cn (Z.W.);
blguo@xidian.edu.cn (B.G.)

* Correspondence: hewp@xidian.edu.cn; Tel.: +86-158-2930-1428

**Abstract:** Superpixel segmentation usually over-segments an image into fragments to extract regional features, thus linking up advanced computer vision tasks. In this work, a novel coarse-to-fine gradient ascent framework is proposed for superpixel-based color image adaptive segmentation. In the first stage, a speeded-up Simple Linear Iterative Clustering (sSLIC) method is adopted to generate uniform superpixels efficiently, which assumes that homogeneous regions preserve high consistence during clustering, consequently, much redundant computation for updating can be avoided. Then a simple criterion is introduced to evaluate the uniformity in each superpixel region, once a superpixel region is under-segmented, an adaptive marker-controlled watershed algorithm processes a finer subdivision. Experimental results show that the framework achieves better performance on detail-rich regions than previous superpixel approaches with satisfactory efficiency.

**Keywords:** adaptive segmentation; superpixel; watershed; coarse-to-fine

## 1. Introduction

Image segmentation has been widely employed in a wide range of computer vision applications, which is essentially a process of dividing an image into several fragments without intersecting. A superpixel [1] is a homogeneity description of texture, color and other features in accordance with visual sense. As the term "superpixel" suggests, it meets the goal of representing an image by perceptually meaningful entitieswhich heavily reduces the number of pixels. This is why superpixels can significantly improve the efficiency of segmentation in practice and become a key preprocessing step for advanced tasks such as video segmentation [2], target tracking [3], object recognition [4], super-resolution [5] and depth estimation [6].

The existing superpixel methods can be generally divided into two categories: graph-based methods and gradient ascent methods. In a graph-based algorithm, superpixels are produced by minimizing a cost function defined over the graph in which each pixel is regarded as a node. Normalized cuts (Ncut) [7] is a representative algorithm based on contour and texture information resulting in regular and compact superpixels, however, it is poor in accuracy and computing efficiency, especially in dealing with large scale images. Felzenszwalb and Huttenlocher [8] propose an efficient graph-based approach through the minimum spanning tree, which shows relative precise adherence to image boundaries, but the procedure is unconscious and the patches are very irregular. Compact superpixels and Constant-intensity superpixels [9], as known as GCa and GCb, are two approaches of a global optimization approach based on [10]. In those frameworks, overlapping image patches are stitched together to generate superpixels where every single pixel belongs to one of the overlapping regions. Superpixels from GCa has the property of uniform compactness with regular shape and size, while GCb performs better in boundary adherence, which shows accurate segmentation precision. However,

these two methods are difficult to adjust parameters and limited in subsequent processing. Entropy Rate Superpixel (ERS) [11] maps one image to an undirected graph consisting of vertices and edges sets, so that a subset of edges is selected to form the resulting graph with directly controlled number of sub-graphs based on entropy rate. Lazy Random Walk (LRW) [12] superpixel segmentation also converts segmentation into graph partition, which iteratively optimizes superpixels by a new energy function and shows well segmentation accuracy, but the time consumption is unsatisfactory for later-stage processing.

Gradient ascent methods, also called clustering-based methods, propose the idea of clustering and iteratively refine the process until it meets a pre-defined criterion. Turbopixels [13] adopts level-set based geometric flow for each seed to generate dense over-segmented and compact superpixels. It combines a curve evolution model for dilation with a skeletonization process for spatial constraint, but sometimes it provides unsatisfactory results in practice. Watershed [14] is a relatively fast segmentation approach based on the topological theory with mathematical morphology. The implementation can be described as a flooding process, it detects the minima of gradients image and pixels at the minima will be flooded. However, the amount of superpixels and their compactness is out of control. Simple Linear Iterative Clustering (SLIC) [15] utilizes local k-means clustering to partial pixels based on color and spatial distance. Compared to many state-of-the-art superpixel methods, SLIC outperforms in several desirable properties for superpixel segmentation, such as the controllability of desired number and compactness through input parameters. Linear Spectral Clustering (LSC) [16] adopts a variant of k-means method to iteratively refine uniformly sampled superpixels similar to SLIC. Whereas it applies a weighted k-means method in the transformed 10-dimensional feature space by kernel function, which further makes an improvement of Ncut [7].

Most superpixel segmentation methods cannot become practical because of their high complexity and memory requirements. In recent years, advanced deep learning applications and excellent structural improvements are proposed to make superpixel algorithms more outstanding. Lv et al. [17] acquire homogeneous change samples from Synthetic Aperture Radar (SAR) images by SLIC, which are then fed to a feature learning method based on the stacked Contractive AutoEncoder (sCAE) to learn the features for change detection. Zhou et al. [18] propose a novel fully supervised scheme for semantic segmentation based on LSC superpixels, which utilizes the advantage of adaptive representation of superpixels context by inferring superpixel-based continuous Conditional Random Field (C-CRF) on features of full resolution. Jia et al. [19] introduce the non-stationarity measure into distance measure and propose nSLIC, which is variable in accordance with local image feature and eliminate the compactness parameter, and eventually improves the visual performance and computing efficiency. To address the segmentation problem of generating structure-sensitive superpixel (SSS) [20], Manifold SLIC (MSLIC) [21] represents the input image as a 2-dimensional manifold, whose area elements are a good measure of content density. SSS are then achieved by computing a Restricted Centroidal Voronoi Tessellation (RCVT) on the manifold, which can be computed with very little cost. What's more, instead of inventing new algorithms, some NVIDIA CUDA-based GPU implementations of state-of-the-art superpixel algorithms are proposed, such as gSLIC [22] and gLSC [23], which modify the original structures thus making them more suitable to parallel deployment and faster-than-real-time application.

It is also worth noting that the watershed runs in $O(N \log N)$, whereas it can be performed with high efficiency empirically. In addition, a critical provision for better segmentation performance is to determine the seed in advance [24]. Wu et.al [25] propose a morphological reconstruction-based approach to identify collection basin markers, and effectively improve the accuracy of froth image segmentation. Hu et.al [26] introduce spatial constraint and edge-preserving to a SLIC-like grid scheme to generate uniform watershed superpixel, which offers controllability on superpixel number and their compactness.

In practice, trade-offs almost always have to be made that should balance some characteristics of segmentation algorithm, so that it can significantly optimize one aspect with slight decrease of another. In this paper, a two-stage image segmentation framework is proposed, which combines two

gradient ascent methods with a coarse-to-fine partition strategy. Firstly, the image is partitioned to regular superpixels by a novel speeded-up SLIC (sSLIC) with emphasis on time efficiency. Based on the calculation of sSLIC, a homogeneity criterion is put forward to define under-segmentation on all sSLIC superpixels. An adaptive marker-controlled watershed algorithm is then proposed to subdivide the misclassified pixels in every heterogeneous superpixel region. Finally, after two-times trade-offs between runtime and accuracy, the framework achieves a better overall performance.

The rest of this paper is organized as follows. Section 2 presents a preliminary on the conventional SLIC and watershed algorithm. Section 3 explicates the proposed two-stage segmentation framework in detail. Qualitative and quantitative analysis are presented in Section 4, Section 5 gives the conclusions.

## 2. Conventional Gradient Ascent Method

### 2.1. SLIC Superpixel Method

The principle of SLIC superpixel is very concise to understand, the overall process contains four major steps: initialization, assignment, updating and post-processing. The algorithm can be described as follows:

- The expected superpixel number $k$ is assigned manually to determine the grid interval $S = \sqrt{N/k}$, where $N$ is the pixel number of the Lab image to be partitioned;
- $k$ initial cluster centers are initialized on the uniform grid in the image plane and represented as a feature vector $C_k = [l_k, a_k, b_k, x_k, y_k]$, where $C_k$ is composed of $C_k^c = [l_k, a_k, b_k]$ in color space and $C_k^s = [x_k, y_k]$ in 2-dimensional space position;
- Each pixel $i$ is assigned a label in accordance with the nearest cluster center $C_k$ based on a distance measure $D(i, C_k)$ as

$$D(i, C_k) = \sqrt{\|C_i^c - C_k^c\|^2 + \rho\|C_i^s - C_k^s\|^2},\qquad(1)$$

where $\rho = 100/S^2$ is a default factor in [15] to normalize color and spatial proximity, and $\|\cdot\|$ represents the Euclidean distance;

- A local k-means method is adopted to adjust the center and the labels of pixels in every $2S \times 2S$ region. This procedure goes until all pixels get new labels and all centers update to $C_k'$ as

$$C_k' = \sum_{i \in \Omega_k} \left[ C_i^c, C_i^s \right] / n_k,\qquad(2)$$

where $\Omega_k$ means the cluster centered at $C_k$, and $n_k$ is the number of pixels in $\Omega_k$. This step is iterated until it reaches a predefined global termination;

- The isolated fragments are merged to a final superpixel $\Omega_k^m = \{\Omega_k, \Omega_{k1}, \cdots, \Omega_{kn}\}$ by region growing method, where $\Omega_{ki}$ indicates a small region unconnected to its cluster but eventually relabeled the same as $\Omega_k$, so that the connectivity among superpixels can be enforced.

### 2.2. Marker-Controlled Watershed Segmentation

Conventional watershed segmentation treats the gradient image as a topographic surface and then floods from the minima based on region-growing. Eventually, the image can be partitioned into catchment basins and watershed lines, which correspond to the homogeneous regions in theory. Once the gradients are distributed irregularly with image noise, it would be submerged by irrelevant boundary.

Meyer [27] introduced a marker extraction strategy to moderate the segmentation result. The extracted markers representing the interior of different objects are regarded as minima of gradient image and suppress all other gradient minima. Then watershed algorithm is used on the modified gradient image for partition optimization. The detailed implementation can be seen in [27].

In this paper, the approach is adopted for subdivision and a new adaptive marker-extraction strategy is proposed based on clustering information.

## 3. Proposed Two-Stage Adaptive Image Segmentation Framework

This section introduces the two-stage segmentation framework in detail. A speeded-up strategy is put forward to reduce the computation redundancy in conventional SLIC, and reconstructs the images by the proposed sSLIC. The under-segmentation regions with heterogeneity are distinguished by a simple criterion, based on which subtle segmentation results can be obtained by an adaptive marker-controlled watershed subdivision in Figure 2.

### 3.1. Speeded-Up simple Linear Iterative Clustering

Figure 1 illustrates the subjective comparison of SLIC and the proposed sSLIC by three images. In general, with the increasing of iterations, the segmentation performance becomes better. In fact, the cluster is updated by the local information of pixels in a certain region, the new center is displaced and more pixels are correctly classified. Achanta [15] indicates that 10 iterations are sufficient for most natural images, and conventional SLIC uses it as a fixed parameter in its open source code. Practically, that work takes up most of the runtime, by statistical analysis, along with the iteration progressing, the proportion of updating step or iteration period also increases. For one $321 \times 481$ image, it even spends half of total time when iteration exceeds 8 times. Besides, it is obvious in the first two rows of Figure 1, the segmentation results change hardly when iterations increase from 5 to 10. Due to indiscriminately global termination criterion by (2) in SLIC, this drawback usually results in redundant revisiting for clusters without large changes. What's more, in many specific applications such as saliency detection, video compression and target recognition, it is reasonable to pay less attention on background, especially in the image border [28].

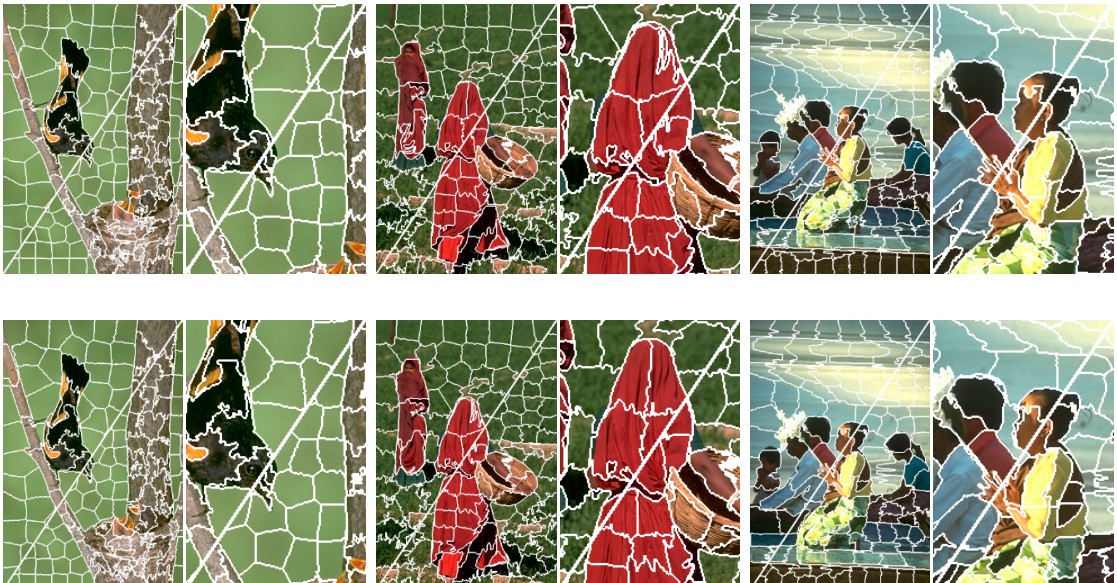

**Figure 1.** *Cont.*

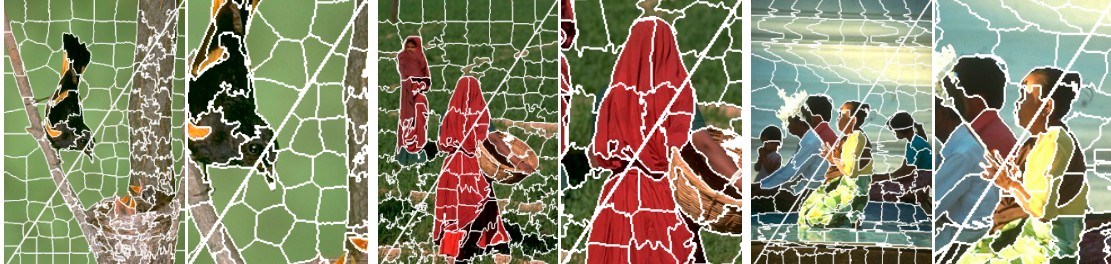

**Figure 1.** Visual comparison of segmentation results. The first two rows display Simple Linear Iterative Clustering (SLIC) superpixels after 5 and 10 iterations respectively. The third row displays speeded-up Simple Linear Iterative Clustering (sSLIC) superpixels. In each image, the expected superpixel number in the upper left and lower right is 100 and 200 respectively. Alternating columns show each segmented image followed by zoom-in performance of the center. See Section 4 for qualitative and quantitative evaluations.

Therefore, the efficiency could be higher if the algorithm is adaptive for local difference. In this work, the deviation of cluster centers in each iteration is obtained as a local interruption criterion to guide the convergence of candidate regions. The only difference is that, in updating step, local k-means method would not be adopted in every search region of clusters, only some with instability repeat to calculate the spatial-color feature for the new center.

In what follows the segmentation algorithm is described in more details. Before the second iteration, each cluster center has moved from its original grid center by initializing to a new position $C'_{k'}$, which can be recorded and described as a feature set $C' = \left\{C'_i\right\}_{i=1}^{k}$. After updating, all elements of $C'$ change in various degree and a new set $C''$ is obtained, and the spatial offset of corresponding $i$th element from $C'$ to $C''$ can be defined as $\Delta C_i^{s\prime} = \|C_i^{s\prime\prime} - C_i^{s\prime}\|$. Then in $\Delta C_i^{s\prime} = \left\{\Delta C_i^{s\prime}\right\}_{i=1}^{k}$, a threshold parameter *TH* is adopted to control the subsequent iterations. If $\Delta C_i^{s\prime} < TH$, cluster centered at $C_i^{s\prime\prime}$ would be evaluated as a homogeneous region, and most pixels tend to belong to the cluster. Therefore, the subsequent iterations become unnecessary, which should be aborted to avoid redundant computation. This modified process is repeated 10 times or all clusters stop updating in advance, which eventually makes a difference among all clusters during updating and effectively avoids a mass of pixels being compared by (1). A visual evaluation of sSLIC superpixel segmentation is shown in the last two rows of Figure 1, it is obvious that segmentation performance dose not degrade badly. A quantitative analysis of speed improvement is described in Section 4.

### 3.2. Adaptive Marker-Controlled Watershed Subdivision

SLIC superpixel segmentation exposes some drawbacks due to its simple framework, not only resulting in redundant eigenvalue computation, but becoming a bottleneck of performance. For example, other than LSC superpixel [16], there is no global image property considered in local-based k-means clustering, which would lead to wrong gathering for some pixels during clustering, and eventually be mislabeled. What's worse, SLIC adopts a split-and-merge post-processing, and produces a large number of heterogeneous regions if isolated fragments aggregate without accurate guidelines [24]. Aimed at these situations, a feasible approach is to find the superpixels without uniformity and then subdivide them in a more precise way.

For efficiently sieving the superpixels mentioned above, as well as keeping the external outliers of SLIC segmentation, a homogeneity criterion is put forward to define under-segmentation. Since almost all superpixels are merged and relabeled after post-processing, along with discarding their local information during clustering and aggregating, the criterion is then divided into two cases for different superpixels:

- If a superpixel is still simply connected without merging neighboring isolated regions (in Figure 2c, they are filled with blue and green), namely $\Omega_k^m = \Omega_k$, the inner difference in Lab color space can be calculated by:

$$d\left(C_i^c\right) = \|C_k^c - C_i^c\|, \tag{3}$$

$$C_i^{\min} = \arg \min_{C_i \in \Omega_k} \left(d\left(C_i^c\right)\right), \tag{4}$$

$$C_i^{\max} = \arg \max_{C_i \in \Omega_k} \left(d\left(C_i^c\right)\right), \tag{5}$$

where $C_i^{\min}$ and $C_i^{\max}$ are a pair of pixels with the minimum and maximum distances from cluster center $\Omega_k$ respectively. The superpixel is considered heterogeneous if:

$$\|C_i^{\min} - C_i^{\max}\| > \varepsilon. \tag{6}$$

- If a superpixel $\Omega_k^m$ merges neighboring isolated regions (cyan and yellow parts in Figure 2c), the mean value of Lab color space is obtained as a region vector $C_{ki}^c = [l_k, a_k, b_k]$ to the neighboring region $\Omega_{ki}$. The sum of inner difference between $C_k^c$ and each $C_{ki}^c$ is computed for determining the heterogeneity of $\Omega_k^m$ if

$$\sum d\left(C_{ki}^c\right) > 2\varepsilon. \tag{7}$$

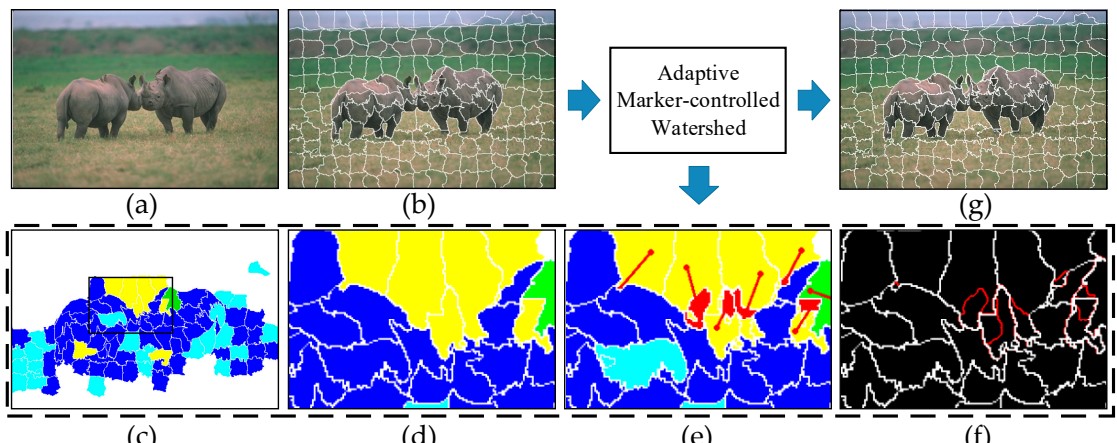

**Figure 2.** The schematic diagram of proposed two-stage image segmentation framework. (**a**) Input image; (**b**) sSLIC Superpixels; (**c**) Under-segmentation classification by homogeneity criterion; (**d**) Zoom-in part of (**c**); (**e**) Adaptive marker extraction from under-segmentation superpixels, each set of markers (solid circle or filled portion) in one superpixel is matched in red; (**f**) Subdivision by watershed, and red outlines are newly emerged boundaries; (**g**) Segmentation result.

In the next step, the adaptive marker-controlled watershed approach is adopted to subdivide the under-segmentation superpixels (green and yellow regions in Figure 2e). In the first case, a set of markers for topographic surface flooding is defined as $C_i^{\min}|C_i^{\max}$ by (4)–(6). Another situation is $C_k|\Omega_{ki}$ , where $C_k$ is the cluster center of $\Omega_k$ and $\Omega_{ki}$ represent(s) the entire partition of isolated region(s) with relative large color difference by (7) and (8):

$$d\left(C_{ki}^c\right) > \frac{\varepsilon}{2}. \tag{8}$$

notice that it is reasonable for the proposed sSLIC to skip subdividing the regions that converge in advance, which nearly all manifest homogeneity and do not need evaluating (white parts in Figure 2e). It is also implied that sSLIC preserves strong homogeneity from SLIC since only a few superpixels are

judged under-segmentation. Moreover, the procedure fully utilizes intermediate computation in SLIC algorithm and introduces the only parameter $\varepsilon$ to control strictness of the criterion (in this paper, $\varepsilon = 3$ is used).

### 3.3. Coarse-to-Fine Segmentation Framework

A major insight of sSLIC from previous work can be generalized as a trade-off between segmentation quality and time efficiency. The proposed local interruption criterion may affect the integrity of spatial context information in global updating more or less while avoiding much distance computation. On the contrary, time saved in that coarse procedure, as well as cues for potential under-segmentation regions, would be available for a finer partition, such as the aforementioned distance-dependent adaptive marker-controlled watershed. As a result, under-segmented superpixels anticipated by the homogeneity criterion are subdivided in a finer level, resulting in better boundary adherence. The proposed coarse-to-fine segmentation framework is summarized in Algorithm 1.

---

**Algorithm 1:** The proposed coarse-to-fine segmentation framework

---

**Input:** the Lab image I, the expected superpixel number k
/* Initialization */
Initialize cluster centers and assign starting labels similar as conventional SLIC.
/* sSLIC Coarse Segmentation */
**if** 1st iteration **then**
   set spatial offset $\Delta C_i^{s'} = \infty$ for each cluster center
   set iteration time $itr = 1$ for each cluster center
**else**
  **repeat**
    **for** each cluster center $C_k$ **do**
      Compute $\Delta C_i^{s'}$.
      **if** $\Delta C_i^{s'} < TH$ **then**
        skip calculating pixels in the cluster centered at $C_i$
      **end if**
      Assign and update superpixel the same as conventional SLIC.
      $itr \leftarrow itr + 1$.
    **end for**
  **until** $itr = 10$ or all pixels are skipped
**end if**
/* Adaptive Marker-controlled Watershed Finer Subdivision */
**for** each sSLIC superpixel **do**
  **if** $itr < 4$ **then**
    compute markers by the homogeneity criterion
    run marker-controlled watershed algorithm in the superpixel region
  **end if**
**end for**

---

## 4. Experiment and Analysis

The experiments are performed on the Berkeley Segmentation Data Set and Benchmarks 500 (BSDS500) [29]. The images for segmentation are all $481 \times 321$ and $321 \times 481$ in size, along with manual ground truth. The proposed framework is compared with watershed and SLIC [15] to prove the effectiveness, as well as LSC [16] to demonstrate the superiority. All algorithms are based on available code with default parameters by the authors except for watershed, which is modified by OpenCV implementation. The homogeneity threshold $TH$ is set to 0.002 times the size of one sSLIC superpixel in expectation. In this paper, all experiments are carried out on an Intel Core i5-6500 PC with a 3.2 GHz CPU and 8G RAM.

*4.1. Visual Comparison and Quantitative Metrics*

Figure 3 provides four results for visual comparison of superpixels obtained by the above-mentioned algorithms. As is intuitively depicted, SLIC, sSLIC and LSC all present relative compact and uniform superpixel apart from watershed. Nevertheless, in order to maintain compactness, SLIC fails in some detail-rich regions such as ear of the rhinoceros and antler of the wapiti. Moreover, the biggest advantage of SLIC is the high efficiency, which may sacrifice some performance, for instance, the content awareness. Therefore, in some regions with weak boundaries, e.g., the head of the leopard, it is difficult for SLIC to attach the actual borders since it merely relies on clustering of local color-spatial features [30].

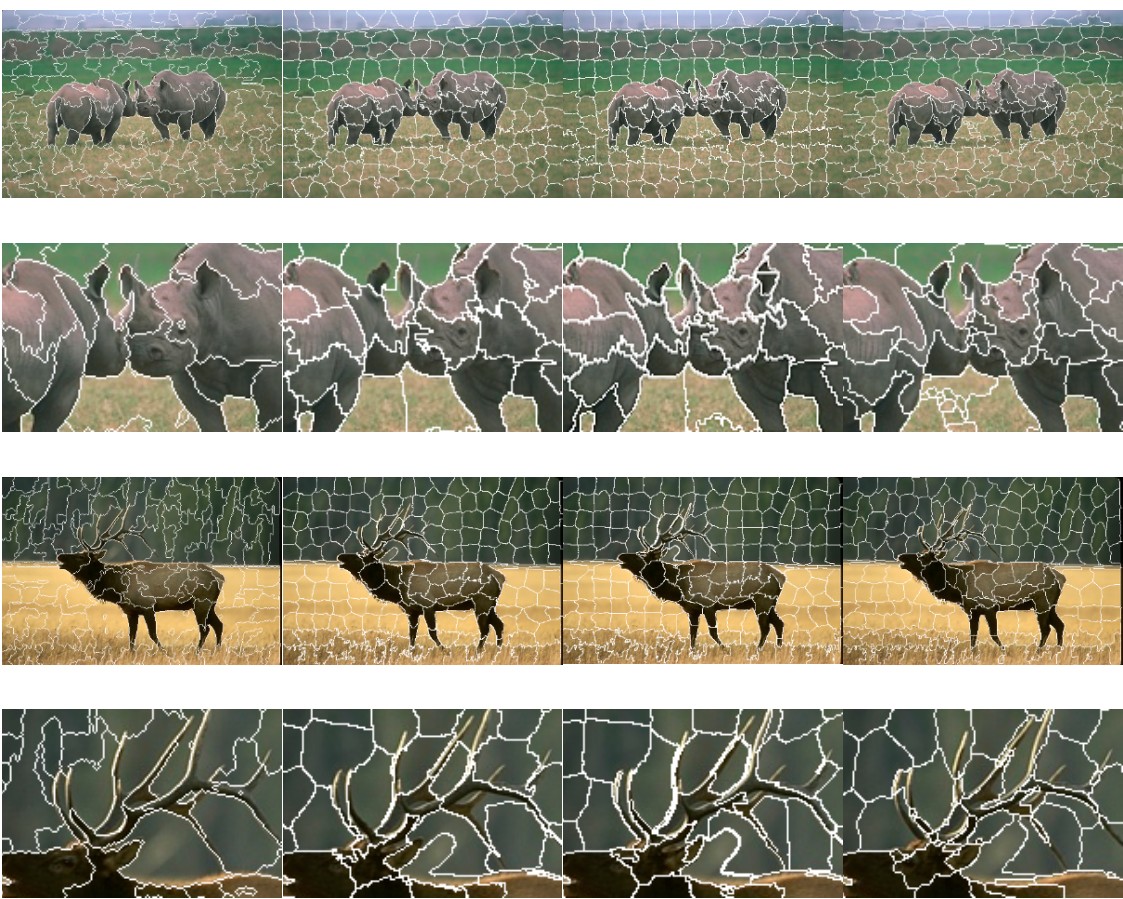

**Figure 3.** *Cont.*

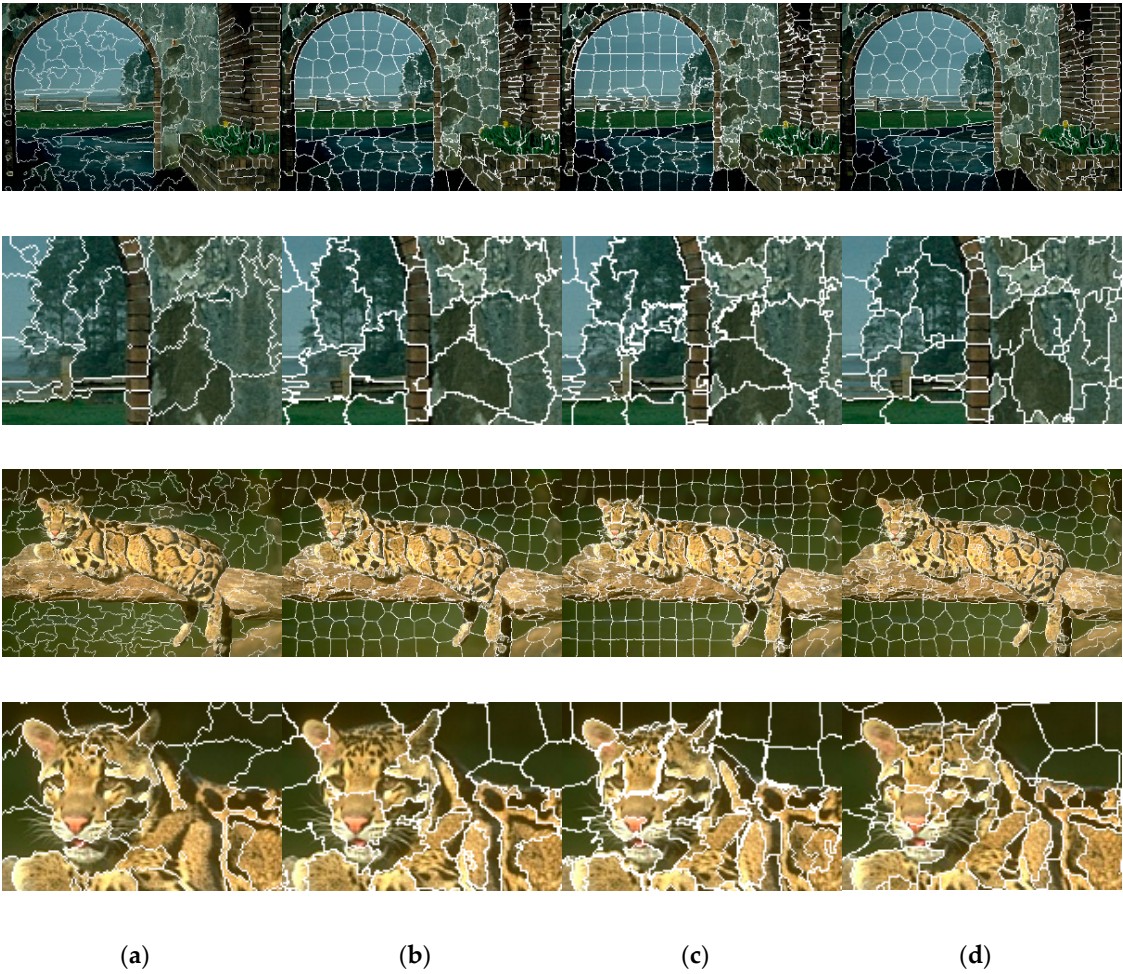

(**a**)　　　　　　　　(**b**)　　　　　　　　(**c**)　　　　　　　　(**d**)

**Figure 3.** Visual comparison among four different methods. (**a**) Result by Watershed; (**b**) Result by SLIC Superpixel; (**c**) Result by the proposed method; (**d**) Result by Linear Spectral Clustering (LSC) Superpixel. Alternating rows show each segmented image followed by local details of each image.

Although the segmentation quality seems undesirable, watershed has two merits well suited for superpixel subdivision. One is the ability to segment patches with any shape, avoiding regular region restriction such as SLIC and LSC. The other is spatial arrangement of the resulting regions by the choice of markers mentioned in Section 3.2, which which gets rid of the constraint of spatial compactness. Therefore, if markers are set in under-segmentation regions with weak boundaries and complicated textures properly, then watershed would perform subtle local treatments and show a more accurate outline detection.

For quantitatively evaluating the performance of boundary adherence, two commonly used evaluation metrics in superpixel segmentation methods are taken into account in this subsection. Specifically, boundary recall (BR) [29] and under-segmentation error (UE) [15] are adopted, with emphasis on edge and region consistency, respectively.

BR measures the degree of ground truth boundaries covered by superpixel boundaries. According to [11], coverage radius is set to 1 pixel. As revealed in Figure 4a, the proposed method outperforms other three comparative methods in a wide range of superpixel size, which is in accordance with the subjective comparison in Figure 3. The superiority of the two-stage framework is owing to utilizing the idea of hierarchical partition by SLIC and Watershed. In fact, during the speeding up clustering process, sSLIC inherits and adopts local k-means method in SLIC, without considering any global constrain strategy differently than LSC (that is why the overall BR of SLIC is smaller than the latter). Besides, the compact property seems easy to degrade in complicated textures regions

constrained by single color-spatial distance. Therefore, while resulting partitions subdivided by watershed are highly irregular in size and shape, they still perform satisfactory boundary adherence. As a result, the following two properties, regularity and perceptual satisfaction sacrifice a little for a better boundary recall.

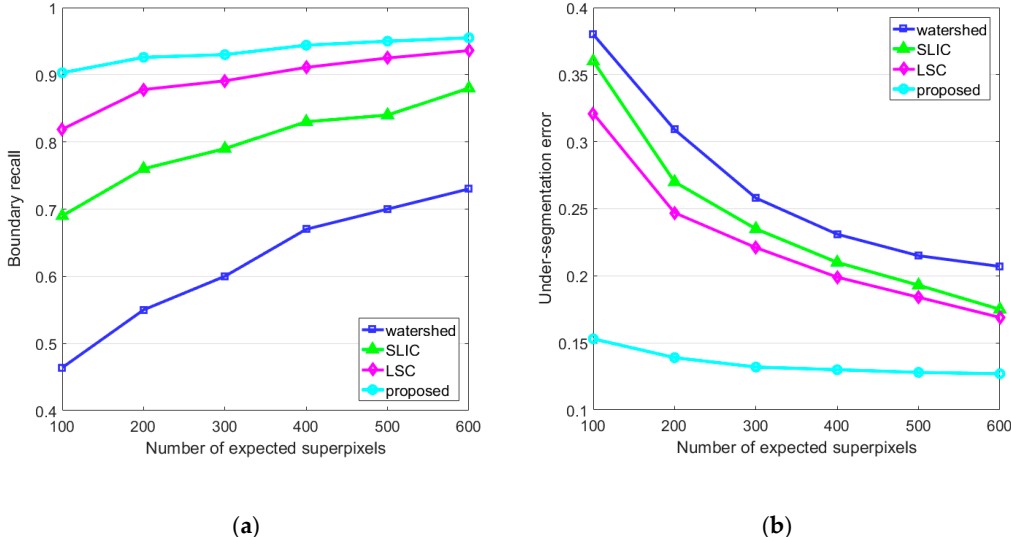

(**a**)            (**b**)

**Figure 4.** Quantitative evaluation of different algorithms in terms of boundary adherence. (**a**) Boundary recall; (**b**) Under-segmentation error.

UE measures the ability of a group of gathered superpixels depicting the ground truth object, which indicates the representation capacity for superpixels as region-based features. The metric follows [15] and sets the tolerance to 5%. As shown in Figure 4b, the proposed method outperforms the three others with the lowest UE in a wide range of superpixel density. Since SLIC holds an acceptable performance on UE, the proposed framework is able to separate almost all superpixels overlapping with parts of an object by ground truth boundaries even further. Eventually, the newly consequent outlines draw a more concrete region of the object. It is worth mentioning that, some under-segmented regions only suffer "dichotomy" into two sub-regions, which is proved experimentally reasonable in practice.

*4.2. Algorithm Complexity and Computational Efficiency*

Since superpixels are often generated to speed up the subsequent visual analysis, the algorithm should performs efficiently in various practical tasks. In the process of generating sSLIC superpixels, clustering regions without apparent changes are excluded from global iteration. For that reason, aggregation computation on a number of pixels can be reduced significantly.

Figure 5 describes the dynamic iteration guided by local interruption criterion. As mentioned in Section 2, the algorithm is modified to adapt local difference, thus it is related to intrinsic image characteristics. Three images with different degrees of smoothness and homogeneity are chosen from BSDS500 to illustrate the improvement. As depicted in Figure 5a, a small target is in front of a single background, which can be regarded one simple image. In that case, almost all superpixels are initialized with uniform information that are completely unnecessary to update. On the other hand, once neighboring superpixels contain intersecting information, there is a relatively long time to reach stabilization. Figure 5d counts the amount of changes of interrupted superpixels during iterating those three images with simple, normal, complex information respectively. In general, an increasing number of superpixels skip clustering during global iterating, which in turn reduces the elapsed time in the next iteration. When all regions stop updating, sSLIC terminates in advance, that is the reason why it could achieve higher efficiency than conventional SLIC.

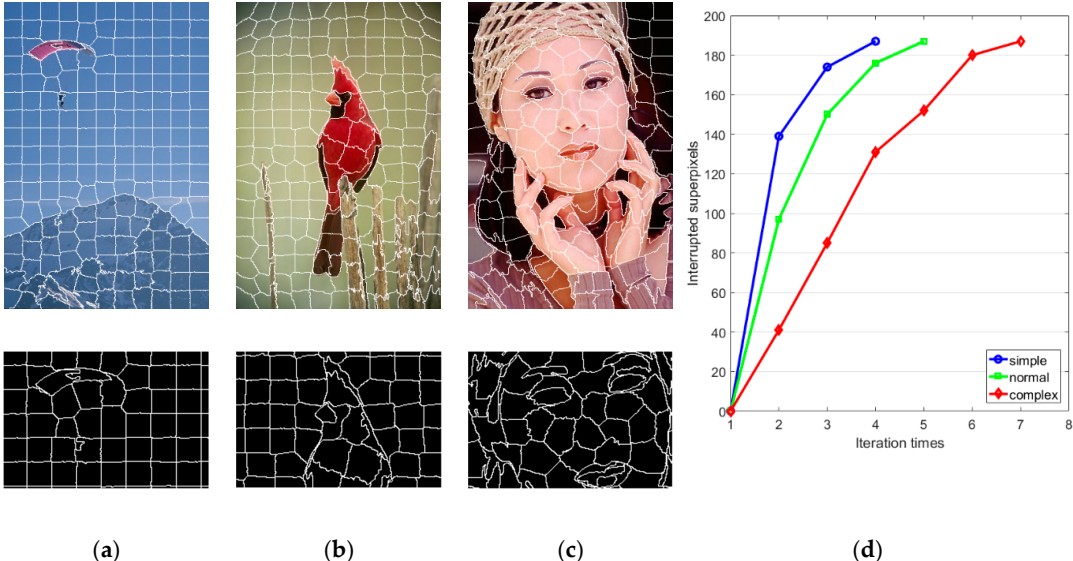

(**a**)                       (**b**)                       (**c**)                       (**d**)

**Figure 5.** Dynamic iteration guided by local interruption criterion in sSLIC processing with different image characteristics. (**a**) Simple image; (**b**) Normal image; (**c**) Complex image; (**d**) The amount of change of interrupted superpixels during iterating.

Figure 6 presents the comparison of runtime on different algorithms. Figure 6a illustrates the performance on BSDS500 dataset with respect to various superpixel sizes. In Figure 6b, plenty of natural images are collected to measure the efficiency in different image scales together with BSDS500. The additions range from $800 \times 600$ to $1600 \times 1200$ with multiple characteristics, and started with approximately 200 pixels in each superpixel on average. As shown in Figure 6, the complexity of SLIC, watershed and LSC is all linearly associated with the number of pixels in images, as well as being irrespective of superpixel size. The proposed method maintains $O(N)$ complexity, which can be high-efficiently performed by sSLIC in global and watershed in small amounts, the remaining time is mainly spent on determining under-segmentation and calculating the markers. But it still has comparable time efficiency to LSC and SLIC.

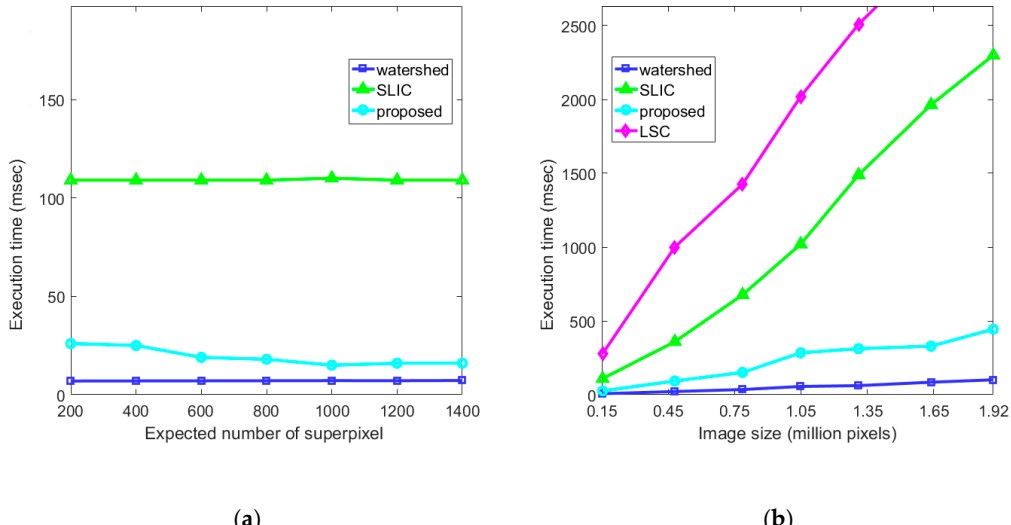

(**a**)                                      (**b**)

**Figure 6.** Comparison of execution time for different algorithm to generate superpixels. (**a**) Time required for superpixels of increasing number (LSC is not plotted due to its relatively slow speed); (**b**) Time required for images of increasing size.

## 5. Conclusions

This paper proposes a two-stage framework for image adaptive segmentation, which combines SLIC and watershed to improve the performance of superpixel. An acceleration strategy is introduced into conventional SLIC, which makes it more efficient to generate superpixels. Then a homogeneity criterion is put forward to define under-segmentation regions in each superpixel, and an adaptive marker-controlled watershed algorithm is adopted to subdivide those regions. After the two procedures in hierarchical order, a more precise segmentation result is obtained. Experimental results demonstrate that the combination makes fully use of clustering information in the framework and improves the performance on detail-rich regions with less time consuming.

Future work will focus on exploring efficient superpixel merging methods since the proposed framework results in much over-segmentation. Moreover, the idea of multi-scale segmentation performance by hierarchical superpixel algorithms will also be considered in future work.

**Author Contributions:** All the authors contributed to this study. W.H.: conceptualization, funding acquisition, project administration, writing of review and editing; C.L.: investigation, writing of the original draft; Y.G. and Z.W.: investigation; B.G.: supervision.

**Funding:** The authors would like to thank the editor and anonymous reviewers for their valuable comments on this paper. This research is supported financially by National Natural Science Foundation of China (Grant No. 51805398), the National Natural Science Foundation of Shaanxi Province (Grant No. 2018JQ5106) and the Fundamental Research Funds for the Central Universities (Grant No. JBX171308).

**Conflicts of Interest:** The authors declare no conflict of interest.

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
