# Peer review of "A Two-Stage Gradient Ascent-Based Superpixel Framework for Adaptive Segmentation"

_applsci, doi:10.3390/app9122421_

Round 1

Reviewer 1 Report

The present submission presents a two-stage super pixel segmentation framework based on a speeded-up simple linear iterative clustering and an adaptive marker-controlled watershed subdivision. Experiments on two benchmark data sets show the improvement of the proposed method compared to more traditional ones. While the proposed approach seems interesting, in my opinion the authors fail to communicate it to the interested audience. The manuscript presents several typos and its reading is difficult. The manuscript should have been out proofread by an English native speaker with a scientific background. In its current state, the work does not meet high quality publication standards and, for this reason, its main contribution is hard to appreciate.

Author Response

Response to Reviewer #1

Comments and Suggestions for Authors:

The present submission presents a two-stage super pixel segmentation framework based on a speeded-up simple linear iterative clustering and an adaptive marker-controlled watershed subdivision. Experiments on two benchmark data sets show the improvement of the proposed method compared to more traditional ones. While the proposed approach seems interesting, in my opinion, the authors fail to communicate it to the interested audience. The manuscript presents several typos and its reading is difficult. The manuscript should have been out proofread by an English native speaker with a scientific background. In its current state, the work does not meet high quality publication standards and, for this reason, its main contribution is hard to appreciate.

Authors:

The authors thank the reviewer for his/her valuable comments and suggestions. Those comments are all constructive and very helpful for revising and improving our paper. We really appreciate that the reviewer commented on our proposed approach as “interesting”. Meanwhile, we agree with the reviewer that “the authors fail to communicate it to the interested audience.” We have considered the valuable comments raised by the reviewer and made changes accordingly in the revised manuscript. Note that changes are marked with blue in the revised manuscript.

The main modifications can be summarized as follows:

(1) We have thoroughly rechecked the manuscript and also invited our English native speaker friends to proofread our work. Lots of grammatical mistakes, typos and inappropriate sentences have been corrected or rewritten in the revised manuscript.

(2) Some inappropriate expressions have been adjusted, which would be more objective and precise.

(3) We have revised the descriptions of Figure 1-3 and Figure 6, which enables this paper easier for readers to understand. Accordingly, the contexts related to all figures has also been changed in the manuscript to better present the contribution and main goal of our paper.

(4) In the introduction section, more background descriptions are presented in the revised manuscript. Moreover, we have added some newly published works in references as a comprehensive review of related work in Section 1. Meanwhile, a few references are also updated to the latest. If the reviewer has more relevant and valuable references related to this topic, please share with the authors and we can add them in the final manuscript.

(5) Notice that the whole section 3 is designed to adequately describe the proposed method. We majorly revised this section on order to present our proposed method to the interested readers. Specifically, a brief paragraph is provided to introduce the idea of our proposed method at the beginning of this section. Notice also that the title of section 3 is also modified to help readers easily catch our proposed method, wherein the contribution is appreciated by the reviewer.

Again, we would like to express our sincere thanks to the reviewer for your constructive comments and suggestions.

Reviewer 2 Report

The article is well written and the results support the claim. I therefore recommend for publication with minor following comment.

1) Figure 1. may be redrawn and provide with more clear description.

Author Response

Response to Reviewer #2

Comments and Suggestions for Authors

The article is well written and the results support the claim. I therefore recommend for publication with minor following comment.

Authors:

The authors thank the reviewer for his/her valuable comments and approval for publication. The comment is constructive and very helpful for revising and improving our paper.

Question 1#

1) Figure 1. may be redrawn and provide with more clear description.

Authors:

Thanks for your kind suggestion. Figure 1 aims to depict the subtle difference of 5-times-iterated-SLIC, 10-times-iterated-SLIC (conventional SLIC) and the proposed sSLIC in following two aspects:

1. 5- and 10-times-iterated SLIC results can be contrasted by the first two rows of segmented images;

2. Conventional SLIC and the proposed sSLIC can be contrasted by the last two rows of segmented images;

As suggested, we have reformatted Figure 1. Since each algorithm is shown by three 321×481 images from BSDS500 with colorful background, it is more visualized to draw the outlines in white.

Besides, we have revised the description of Figure 1 easier to understand.

Accordingly, the contexts related to Figure 1 has been changed in the manuscript, so that it can introduce the advantage of sSLIC more convincingly.

Reviewer 3 Report

Overall, I think that there are merits to this submission which make it a promising candidate for inclusion in Applied Sciences.  The problem is relevant, the approach described is sufficiently novel and sound, and the experimental results are convincing in scope and methodology for the type of challenge involved.  Mathematical content is also clear and complete, which is good

The biggest weakness of the manuscript lies in its presentation and specifically numerous linguistic issues.  There are many awkward phrases which throw off the reader, make some text difficult to read, etc.  Here are some examples:

- "merges unconsciously" (algorithms are not conscious)

- "representing an image accurately using a smaller number of pixels" (smaller than what?  the original number of pixels?  of course not)

- "in image segmentation task" (bad grammar)

There are many many examples of this sort, this is far from an exhaustive list.  The authors should carefully re-read the entire paper and make corrections.

Then there are extreme overstatements, such as calling SLIC "fundamentally flawed".  Just because an algorithm can be improved, that does not mean that it is "fundamentally flawed".  This is inappropriate.

Lastly, the list of references is a bit outdated and the authors should contextualize the importance of the problem by referencing some examples of recent work that relies on superpixels:

- "Super-resolution of hyperspectral image via superpixel-based sparse representation" (2018)

- "Automatic semantic labelling of images by their content using non-parametric Bayesian machine learning and image search using synthetically generated image collages" (2018)

- "Accurate light field depth estimation with superpixel regularization over partially occluded regions" (2018)

Author Response

Response to Reviewer #3

Comments and Suggestions for Authors

Overall, I think that there are merits to this submission which make it a promising candidate for inclusion in Applied Sciences.  The problem is relevant, the approach described is sufficiently novel and sound, and the experimental results are convincing in scope and methodology for the type of challenge involved.  Mathematical content is also clear and complete, which is good.

The biggest weakness of the manuscript lies in its presentation and specifically numerous linguistic issues.  There are many awkward phrases which throw off the reader, make some text difficult to read, etc.  Here are some examples:

Authors:

The authors thank the reviewer for his/her valuable comments and approval for publication. The comment is constructive and very helpful for revising and improving the paper.

Question 1#

- "merges unconsciously" (algorithms are not conscious)

Authors:

Thanks for your kind reminder. We agree with the reviewer and we have noted this in the revised manuscript. The newly added sentence is shown below (L185-187):

“What's worse, SLIC puts a split-and-merge post-processing, and it would produce large number of heterogeneous regions if isolated fragments aggregates without accurate guidelines [24].”

Question 2#

- "representing an image accurately using a smaller number of pixels" (smaller than what?  the original number of pixels?  of course not)

Authors:

Thank you for pointing out the error. We agree with the reviewer’s comments. It has been corrected in the revised manuscript (L26-27)

“As the term “superpixel” suggests, it meets the goal of representing an image by perceptually meaningful entities which heavily reduces the number of pixels.”

Question 3#

- "in image segmentation task" (bad grammar)

Thank you for correcting the mistake in grammar. The last sentence in abstract has been modified (L18-19)

“Experimental results show that the framework achieves better performance on detail-rich regions than previous superpixel approaches with satisfactory efficiency.”

Question 4#

There are many examples of this sort, this is far from an exhaustive list.  The authors should carefully re-read the entire paper and make corrections.

Thank you for pointing out these errors. We have thoroughly rechecked the manuscript and also invited our English native speaker friends to proofread our work. Lots of grammatical mistakes, typos and inappropriate sentences have been corrected or rewritten in the revised manuscript.

Again, we would like to express our sincere thanks to the reviewer for your constructive comments.

Question 5#

Then there are extreme overstatements, such as calling SLIC "fundamentally flawed".  Just because an algorithm can be improved, that does not mean that it is "fundamentally flawed".  This is inappropriate.

Thanks for your valuable suggestion. As suggested, some words and sentences have been adjusted, such as (L181-182)

“SLIC superpixel segmentation exists some drawbacks due to its simple framework, not only resulting in redundant eigenvalue computation, but becoming the bottleneck of accuracy.”

We believe the expressions of the revised manuscript could be more objective and precise.

Question 6#

Lastly, the list of references is a bit outdated and the authors should contextualize the importance of the problem by referencing some examples of recent work that relies on superpixels:

- "Super-resolution of hyperspectral image via superpixel-based sparse representation" (2018)

- "Automatic semantic labelling of images by their content using non-parametric Bayesian machine learning and image search using synthetically generated image collages" (2018)

- "Accurate light field depth estimation with superpixel regularization over partially occluded regions" (2018)

Authors:

Thanks for your kind suggestion. As suggested, we have added and commented on abovementioned valuable references and some other newly published work as a comprehensive review of related work in Section 1. Meanwhile, a few references are also updated to the latest.

Round 2

Reviewer 1 Report

The authors have made substantial effort in improving the quality of the presentation, which now meets a high standard. I only suggest to try to not break figures and captions in multiple pages. 

Reviewer 3 Report

The paper is much improved so I am happy to recommend acceptance.